# Spatial Convergence of Carbon Productivity: Theoretical Analysis and Chinese Experience

**DOI:** 10.3390/ijerph19084606

**Published:** 2022-04-11

**Authors:** Meng Sun, Yue Zhang, Yaqi Hu, Jiayi Zhang

**Affiliations:** 1Center for Northeast Asian Studies, Jilin University, Changchun 130012, China; sunmeng_meng@jlu.edu.cn; 2Northeast Asian Studies College, Jilin University, Changchun 130012, China; zhang_yue20@mails.jlu.edu.cn (Y.Z.); jiayiz18@mails.jlu.edu.cn (J.Z.)

**Keywords:** carbon productivity, convergence hypothesis, fixed effect, spatial effect

## Abstract

Based on the neoclassical framework, we propose the convergence hypothesis of carbon productivity under sustainable growth and prove the different effects of knowledge spillover and technology diffusion on convergence. The convergence hypothesis is tested using China’s provincial spatial dynamic panel data from 1995 to 2019. The results show that China’s provincial carbon productivity has conditional convergence and club convergence characteristics. The convergence speed of dynamic panel regression estimation is greater than that of cross-sectional regression. The convergence rate of dynamic spatial panel regression estimation is faster depending on the spatial spillover difference between the two technologies. In the early stage, the provincial spatial dependence of China’s carbon productivity is mainly knowledge spillover, and the convergence rate is lower than that of the closed economy. Over the past decade, the spatial spillover, dominated by low-carbon technology diffusion, has become the dominant force. The convergence rate is significantly faster than that of a non-spatial-dependent economy. In addition, the mechanism test found that the development of energy efficiency dominates the spatial transfer of technology, so the overall convergence of carbon productivity in China mainly comes from the apparent convergence of energy efficiency in provinces and cities. Our conclusion provides a new reference for the emission reduction actions of countries worldwide because the spatial knowledge spillover carried by capital flows is not conducive to the pursuit of carbon productivity in less developed regions. On the contrary, the dissemination and diffusion of low-carbon technologies can significantly reduce carbon equivalent input in the production process, accelerating the pursuit of developing countries or regions.

## 1. Introduction

Global warming leads to a rapid increase in the probability of extreme climate events, and sustainable economic and social development faces various risks. The main reason for global warming is the greenhouse gas emissions generated by human economic activities [1]. In 2020, the World Meteorological Organization (WMO) released the ‘Greenhouse Gas Bulletin’, which shows that the radiation intensity caused by greenhouse gases in 2019 increased by approximately 45% compared with 1990, and carbon dioxide accounted for approximately 80% of the increase. Global politicians made the proposal to control carbon dioxide emissions to adapt to global climate change [2]. As the largest developing country, and the most significant carbon emission country, China actively participates in global governance and responds to climate change and promises to reduce carbon dioxide emissions per unit of Gross Domestic Product (GDP) by between 60 and 65% by 2030 compared with 2005.

However, there is still a significant gap between the living standards of Chinese residents and developed countries, and there is a strong demand for development [3]. Coordinating the relationship between carbon emission reduction and economic development is an urgent problem for governments to address.

It is generally believed that the motivation of economic growth is to increase input factors and improve productivity. However, with the increase of input factors, the marginal benefit decreases. Thus, the source of long-term economic growth are productivity gains [4]. Carbon productivity refers to a region that provides products and services to meet human needs with fewer carbon emissions, i.e., the ratio of production to carbon emissions is an important indicator of sustainable regional economic development [5]. Carbon productivity is based on economic growth and carbon emission reduction. With the increase in carbon productivity, the economy may increase energy consumption and, subsequently, carbon emissions will increase, which is known as the ‘rebound effect’ [6,7]. However, carbon productivity can significantly inhibit the excessive growth of this emission increment, so carbon productivity is still essential for a low-carbon economy in developing countries [8,9]. It should be noted that there are gaps in resource endowments, the industrial structure, and the economic base in different regions, and there are differences in the spatial distribution of carbon productivity. Is this difference likely to converge over time? At the same time, inter-regional linkages continue to strengthen and the emergence of economic agglomeration and carbon productivity agglomeration have also emerged. Different regions will affect the convergence of carbon productivity through spatial effects such as capital, labor, and technology spillovers. ‘Efficient reproductive effects’ of carbon productivity allow regions to learn from each other and improve carbon productivity, i.e., when carbon productivity in adjacent regions increases, the region will improve its carbon productivity by imitating its technology and management practices [10]. The convergence of carbon productivity has important practical significance for the government to formulate relevant environmental policies, develop a low-carbon economy, and promote regional low-carbon cooperation. Given convergence, the existing literature includes studies about energy intensity convergence [11], carbon emission convergence [12], and total factor productivity convergence [13]. However, they are all based on the traditional Solow model, lacking theoretical support in the field of spatial convergence, and the empirical results cannot be reasonably explained.

Based on the above background, we try to expand on the following three aspects. Firstly, we try to construct a theoretical framework containing different spatial spillover effects from the perspective of neoclassical theory, which provides theoretical support for the carbon convergence hypothesis. Secondly, we use Chinese provincial data and dynamic spatial panel data (SDPD) method to verify the impact of different forms of technology spillover on carbon productivity convergence and provide a Chinese experience and case. Finally, our research also provides a new decision-making basis for supporting developing countries’ energy conservation and emission reduction paths, which has essential practical significance for global climate governance.

In this paper, the following sections are arranged: Section 2 summarizes the empirical literature on carbon productivity convergence; Section 3 introduces the convergence hypothesis and estimation methods; Section 4 introduces our research results in detail using previous data; Section 5 concludes.

## 2. Literature Review

### 2.1. Economic Convergence

The neoclassical growth model believes that the economy has a ‘steady state’ and ‘conditional convergence’; that is, under the assumption of diminishing returns to the scale of input factors, economic growth will eventually reach equilibrium [14]. Economists have frequently discussed whether there is economic convergence in a region’s economic growth. Many scholars use different regional sample data to prove that the region with lower per capita income has faster economic growth than the region with higher per capita income [15,16,17]. The convergence of regional economic growth can be divided into absolute convergence, relative convergence, and club convergence. However, some scholars believe that there is no ‘steady state’ in the economy, changes in initial conditions will have a long-term impact on the economy, and there is no mechanism to ensure that economies tend to ‘converge’ [18,19,20]. Romer improves the endogenous growth model and thinks that knowledge spillover produces economies of scale, making developed countries have higher per capita output [21]. Lucas uses the optimal technological progress model, which assumes that the renewable capital’s returns remain unchanged, and concludes that the per capita output growth rate is independent of the initial per capita output level [22]. Therefore, after relaxing the hypothesis of diminishing marginal returns to capital, it is impossible to obtain the economic growth convergence. However, Bloom et al. (2002) improved the technology diffusion model, considering that the technology diffusion utility shortens the regional technology gap, which makes the economy that of conditional convergence [23].

### 2.2. Environmental Convergence

With the deepening of research, the convergence hypothesis extends to other fields, such as energy efficiency, environmental quality, financial development, and so on [24,25,26]. The sustainable development of the economy has always been the focus of global attention. According to the Environmental Kuznets Curve (EKC), there is an inverted ‘U’ relationship between income and environmental quality [27]. Will economic convergence make the environment converge? Scholars have conducted in-depth studies on this issue [28]. Some scholars have incorporated environmental pollution into the Solow model and found that countries with low environmental efficiency/regions catch up with high environmental efficiency, which verifies the environmental convergence hypothesis [29,30,31,32]. At the same time, scholars’ studies on the externality of environmental pollution have found that environmental efficiency has a spatial spillover effect [33]. Some scholars use Chinese data to find that the convergence rate of carbon productivity in the spatial panel model is higher than that in the non-spatial panel model, which verifies that resource concentration externalities play an important role in improving carbon productivity, narrowing regional disparities, and achieving sustainable growth convergence [5,12]. Economic development brings various resources and environmental problems, especially global warming. Governments are committed to energy conservation and emission reduction, and sustainable growth convergence under carbon emission constraints, that is, carbon productivity convergence, has attracted widespread attention [10,34].

### 2.3. Carbon Productivity Convergence

Scholars mainly focus on the convergence analysis of carbon intensity and carbon emissions and seldom study carbon productivity [35,36]. Scholars use empirical data to analyze the convergence of construction [37], manufacturing, industry, and energy industries [38]. Regional carbon productivity convergence has also received attention. Dong et al. (2013) studied the convergence of regional carbon productivity in China. They found a convergence trend in Chinese carbon productivity, but there is a gap in the convergence rate of regional carbon productivity [39]. Shen et al. (2021) analyzed the convergence of Chinese carbon productivity from urban agglomeration and found that Chinese carbon productivity showed noticeable stickiness and spatial dependence in adjacent areas [40]. Scholars also discussed the influencing factors of carbon productivity. They found that carbon productivity is affected by socio-economic, policy, and energy factors. Technological progress can increase carbon productivity and reduce regional disparities in carbon productivity. With the wide application of space measurement technology, more and more literature strengthens the research on the spillover effect when analyzing the factors affecting carbon productivity. Wu et al. (2021) analyzed 17 cities in Central and Western China. They found that the industrial development and urbanization patterns affecting carbon productivity are homogeneous and mutually imitated and have apparent spatial spillover effects [41]. Zheng et al. (2020) also proved the multiple effects of the economic development level, the industrial structure, and urbanization on carbon productivity [42].

It is worth noting that, due to the complexity of carbon emissions, there is no unified theoretical model and research paradigm. The existing research mainly focuses on carbon emission convergence, per capita carbon emission convergence, and carbon emission intensity convergence [9], which lacks theoretical support and is purely an empirical study. Therefore, from the perspective of carbon productivity, we try to treat carbon equivalent emissions as input factors, reflect the meaning of sustainable growth, and further explore whether there is a convergence hypothesis as traditional economic growth to make up for the lack of existing research.

## 3. Convergence Hypothesis and Estimation Method

### 3.1. Spatial Solow Model

#### 3.1.1. Spatial Spillover and Production Function

The generalized Solow model usually uses empirical norms with spatial spillover economic growth and regional convergence effects. We also consider two forms of technology spillover, so the Cobb–Douglas production function with constant returns to scale is set as follows
(1)$$Q_{it}=B_{it}K_{it}^{\alpha }{\left(A_{it}C_{it}\right)}^{1-\alpha },\;0<\alpha <1$$
where Q is the output level, K is the capital level, C is the carbon emission level, B is the Hicks neutral technology level, A is the low-carbon enhanced technology level, and i and t are the subscript sum which represents the above variables of a given region in a certain period α and is the output share of capital. Let k=K/(AC) and q=Q/(AC) denote the adequate capital and output per unit carbon element, respectively, then a compact production function $q_{it}=B_{it}k_{it}^{\alpha }$ can be obtained.

Referring to the neoclassical growth model, assuming that $A_{it}$ and $C_{it}$ are exogenous rates g and p, there are:(2)$$C_{it}=C_{i0}e^{pt}$$
(3)$$A_{it}=A_{i0}e^{gt}$$

This paper introduces spatial correlation from the perspectives of knowledge spillover and technology diffusion, so technological progress in one region can have spillover effects on other regions [43,44]. We specify the technical level to:(4)$$B_{it}=\prod_{j\neq i}k_{jt}^{\gamma w_{ij}}$$
(5)$$A_{it}=A_{i0}e^{gt}\prod_{j\neq i}A_{j}^{\rho w_{ij}}$$

The above function describes that the technological level of the region i is determined by three factors: Firstly, assuming that the technological level of Hicks $B_{it}$ depends on the knowledge spillover effect of factor flow, it is expressed as the weighted average of adequate capital of three units of carbon in j(j≠i) regions. Secondly, it represents the low-carbon enhanced technology level. It is assumed that $A_{it}$ has partial Solow exogeneity and remains unchanged between regions, that is $e^{gt}$. Finally, we assume that $A_{it}$ has a traditional technology diffusion effect, which shows the weighted average of technology level in other regions. $A_{i0}$ is the initial level of low-carbon technology in a certain region, and γ and ρ represent the interdependence between unit effective capital and low-carbon technology level between regions. The element $w_{ij}$ of the spatial weight matrix W defines the adjacent structure of the technology spillover from other regions to the i region, satisfying the constraints to exclude their own impact. We normalize the row of W, namely, $\sum_{j\neq i}w_{ij}=1$, so that all weights are between 0 and 1. As a result, the higher the proportion of capital and carbon elements in efficiency units, the more advanced low-carbon technologies, the stronger the regional spillover effect. The more advanced the nearby areas, the higher the possibility of learning new technologies.

We can write formula (5) into the following form:(6)$$\ln A_{t}={\left(I-\rho W\right)}^{-1}\ln A_{0}+\frac{gt}{1-\rho }l$$

Among them, $A_{t}$ is the column vector composed of a certain period of low-carbon technology level in each region, $A_{0}$ is the column vector of the initial low-carbon technology level in each region, I is the unit matrix, and l is the unit row vector with all elements of 1. Throughout the process, we represent matrices or vectors in bold letters. The further derivation of time is:(7)$$\frac{{\dot{A}}_{it}}{A_{it}}=\frac{g}{1-\rho }$$

The spillover effect makes the progress of low-carbon technology faster. We allow the iteration of the initial technical level $A_{i0}$, which has been described in detail in the literature. This does not change the spatial spillover effect of low-carbon technologies [45].

#### 3.1.2. Steady Equilibrium and Convergence Analysis

Assuming that the exogenous savings rate is $s_{i}$, the unified depreciation rate is δ, and the change of capital stock is $\Delta K=s_{i}Q_{it}-\delta K_{it}$. Then, according to the chain rule, we can get the dynamic equation of the ratio of efficiency unit capital and carbon elements:(8)$${\dot{k}}_{it}=s_{i}q_{it}-\left(p+\frac{g}{1-\rho }+\delta \right)k_{it}$$

Since the economy is in a steady-state ${\dot{k}}_{it}=0$, the steady-state solution of the capital and output of efficiency unit can be obtained:(9)$$k_{i}^{*}={\left(\frac{s_{i}B_{i}^{*}}{p+g/\left(1-\rho \right)+\delta }\right)}^{1/\left(1-\alpha \right)}$$
(10)$$q_{i}^{*}=B_{i}^{*}{\left(\frac{s_{i}B_{i}^{*}}{p+g/\left(1-\rho \right)+\delta }\right)}^{\alpha /\left(1-\alpha \right)}$$

The matrix form of the steady-state logarithmic equation can be further sorted out by the inclusion of (4):(11)$$\ln k^{*}=\frac{1}{1-\alpha }{\left(I-\frac{\gamma }{1-\alpha }W\right)}^{-1}\ln\frac{s}{p+g/\left(1-\rho \right)+\delta }$$
(12)$$\ln q^{*}=\frac{1}{1-\alpha }\left(\alpha I-\gamma W\right){\left(I-\frac{\gamma }{1-\alpha }W\right)}^{-1}\ln\frac{s}{p+g/\left(1-\rho \right)+\delta }$$

Here $k^{*}$ and $q^{*}$ is the steady-state capital and output vector, and s is the savings rate vector. When the spillover parameter satisfies 0<γ<1−α and (I−[γ/(1−α)])W is a reversible matrix, we can obtain a balanced growth path. The equation can be simplified if savings rates are the same in all regions ${\left(I-\left[\gamma /\left(1-\alpha \right)\right]W\right)}^{-1}l=\left(1-\alpha \right)/\left(1-\alpha -\gamma \right)$. It is not difficult to see that the two technological advances have different spatial spillover effects. The diffusion of low-carbon technology will accelerate the overall technological growth rate and reduce the steady-state capital level because the practical carbon factor input in the production function increases. Hicks’s neutral technology spillovers further expand output, raising steady-state capital levels. When γ=ρ=0, there is no spatial spillover effect between regions, which is the same as the steady-state level of the traditional Solow model.

In order to obtain the output dynamics of practical unit carbon elements, we first write the production function into a matrix form of $\ln q_{t}=\left(\alpha I+\gamma W\right)\ln k_{t}$, and then logarithmically linearize it around its steady-state:(13)$$\begin{array}{ll} \frac{\partial \ln q_{t}}{\partial t} & =\left(\alpha I+\gamma W\right)\frac{\partial \ln k_{t}}{\partial t}\\ & \approx -\left(1-\alpha \right)\left(p+\frac{g}{1-\rho }+\delta \right)\left(I-\frac{\gamma }{1-\alpha }W\right)\left(\alpha I+\gamma W\right)\left(\ln k_{t}-\ln k^{*}\right)\\ & =-\left(1-\alpha \right)\left(p+\frac{g}{1-\rho }+\delta \right)\left(I-\frac{\gamma }{1-\alpha }W\right)\left(\ln q_{t}-\ln q^{*}\right)\\ & :=-\Phi \left(\ln q_{t}-\ln q^{*}\right) \end{array}$$

Since the eigenvalue of W is a real number and satisfies $|\lambda_{i}|\leq 1$, and the corresponding standardized eigenvector matrix is P, then it must be diagonalized to $P^{-1}WP=Diag\left(\lambda_{i}\right)$. Further analysis shows that $P^{-1}\Phi P=\left(p+g/\left(1-\rho \right)+\delta \right)Diag\left(1-\alpha -\gamma \lambda_{i}\right)$. Because 0<λ<1−α, the eigenvalues of the matrix Φ are positive real numbers. Therefore, the differential equations of the above spatial Solow model are also stable, which means that the low-carbon economy system will converge to a unique steady state. The solution of the equations can be given by the following equation [46]:(14)$$\ln q_{t}=e^{-\Phi t}\ln q_{0}+\left(I-e^{-\Phi t}\right)\ln q^{*}$$

Here $e^{-\Phi t}=I-\Phi t+\Phi^{2}\left(t^{2}/2!\right)-\cdots$. From the eigenvalue $\left(p+g/\left(1-\rho \right)+\delta \right)\left(1-\alpha -\gamma \lambda_{i}\right)$ of Φ, it can be seen that low-carbon technology spillover improves the convergence speed, and neutral technology spillover reduces the convergence speed because the former accelerates the overall technology growth rate, while the latter slows down the decline of marginal output.

### 3.2. Estimated Equation

Defines carbon productivity as y=Q/C=Aq and rewrites it as matrix form $\ln y_{t}=\ln A_{t}+\ln q_{t}$, and then brings in Equations (6), (12), and (14), which are left multiplied by the matrix (I−ρW) on both sides:(15)$$\begin{array}{ll} \ln y_{t} & =\rho W\ln y_{t}+e^{-\Phi t}\ln y_{0}-\rho e^{-\Phi t}W\ln y_{0}+\left(I-e^{-\Phi t}\right)\ln A_{0}+gt\cdot l\\ & +\frac{1}{1-\alpha }\left(I-e^{-\Phi t}\right)\left(\alpha I+\gamma W\right){\left(I-\frac{\gamma }{1-\alpha }W\right)}^{-1}\ln\frac{s}{p+g/\left(1-\rho \right)+\delta }\\ & -\frac{1}{1-\alpha }\left(I-e^{-\Phi t}\right)\left(\alpha I+\gamma W\right){\left(I-\frac{\gamma }{1-\alpha }W\right)}^{-1}\rho W\ln\frac{s}{p+g/\left(1-\rho \right)+\delta } \end{array}$$

Using the assumption that the exogenous savings rate is equal to simplify the above equation, we can obtain an empirical system consistent with the theoretical system. Therefore, we have the following estimation equation [47]:(16)$$\ln y_{t}=\rho W\ln y_{t}+\beta \ln y_{t-1}+\theta W\ln y_{t-1}+c_{n}+g_{t}+v_{nt}$$

Here λ=(p+g/(1−ρ)+δ)(1−α−γ) is set, then the time lag parameter in the above equation is $\beta =e^{-\Phi t}\cdot l=e^{-\lambda }$, and the time-space lag parameter is θ=−ρβ. $c_{n}=\left(1-e^{-\lambda }\right)\left(\ln A_{0}+\frac{\alpha +\gamma }{1-\alpha -\gamma }\right)\ln\frac{s}{p+g/\left(1-\eta \right)+\delta }-\rho \frac{\alpha +\gamma }{1-\alpha -\gamma }W\ln\frac{s}{p+g/\left(1-\eta \right)+\delta }$ is the individual effect, in addition to the initial technical level and savings rate, and can also include other individual characteristics. $g_{t}$ is the time effect, and the remaining part $v_{nt}={\left(v_{1t},v_{2t},\cdots ,v_{nt}\right)}^{’}$ is the temporary error term satisfying the assumption of i.i.d. It is not difficult to see that the estimation Equation (16) is consistent with the spatial Durbin model (SDM), and the time lag term on the right side of the equation also satisfies the dynamic panel model. Therefore, we use the quasi-maximum likelihood (QML) method of spatial dynamic panel data to estimate the model implied in the above theory. Since Equation (16) includes time lag term, spatial lag term, and spatial-temporal lag term simultaneously, we will mainly choose between the SDM model of dynamic panel data and the spatial lag model (SLM). Of course, we also report the cross-sectional regression results for comparison [48]. In order to ensure that the empirical results are compatible with more general settings, we do not impose parameter limitations (θ=−ρβ) in the estimation.

When the spatial spillover parameters are all 0, Equation (16) has the same form as the Solow model in the textbook. However, due to the spatial spillover effect, the growth of carbon productivity in a region depends not only on its initial level but also on the initial level of neighboring regions. Individual characteristics such as savings rate also have a spatial spillover effect. The regions with high external connectivity or proximity benefit more from the spatial spillover effect, and the corresponding carbon productivity growth will be more obvious. There are significant differences in the direction of the two types of technology spillover on the convergence of carbon productivity. The spatial spillover of low-carbon technology will accelerate the convergence, and the spatial spillover of neutral technology will reduce the convergence rate. The overall direction of the effect depends mainly on the difference in the spatial spillover intensity of the two types of technology.

### 3.3. Data and Variables Description

This paper adopts the panel data of 30 provinces and cities in China from 1995 to 2019, and Hong Kong, Macao, Taiwan, and Xizang are not included due to the lack of data. The GDP, energy consumption, and fixed asset investment of each province and city can be directly obtained from the National Bureau of Statistics (NBS) website. However, the database lacks any information on provincial carbon emissions. Under the assumption of carbon balance, the carbon content in the supply of raw coal, crude oil, and natural gas is equal to that in the total consumption of other fossil fuels. Therefore, this paper uses the reference method given by the Intergovernmental Panel on Climate Change (IPCC) in 2006 to calculate the carbon equivalent emissions from the combustion of primary fossil fuels, namely:(17)$$C_{t}=\sum_{i=1}^{3}AD_{i}\times NCV_{i}\times CE_{i}\times O_{i}$$

Among them, $C_{t}$ is the carbon emissions generated by fossil energy consumption in a specific province in the t year; $AD_{i}$ is the physical consumption of the i type of primary energy; $NCV_{i}$ is the average low calorific value of the corresponding energy; $CE_{i}$ is the carbon content per unit calorific value of the corresponding energy; $O_{i}$ is the corresponding oxidation efficiency. The average low calorific value data were extracted from Chinese ‘General Principles for Comprehensive Energy Consumption Calculation’ (GB/T2589-2020), and the carbon content and oxidation rate were derived from the recommended values of ‘Provincial Guidelines for the Preparation of Greenhouse Gas Inventories’ (UNDP Climate 1041). Considering the differences in the energy quality and combustion technology among regions, we use the total annual standard coal energy consumption of each province and city recently updated by NBS to adjust the heating factor in Equation (17).

Carbon productivity is calculated by the ratio of GDP to carbon equivalent emissions, and the GDP of each province is adjusted to the comparable price in 2010. The year selection of the base period has little effect on the result analysis. Considering that Chinese economic growth rate from 2011–2019 is significantly lower than from 1995–2010, this paper chooses 2010 as the base period. This paper further controls the individual characteristics of the ratio of savings rate to effective depreciation rate, called effective investment rate. Among them, the saving rate index is measured by the proportion of total investment in fixed assets to GDP, and the depreciation rate of fixed assets is assumed to be 5%. Thus, the dependent variable carbon productivity in the regression model is denoted as lny, the influential investment rate is denoted as lns, and w is used to represent the row standardization matrix of geographical adjacency weight (adjacent to Hainan and Guangdong). In addition, the mechanism test later involves two variables, energy consumption per unit of carbon emissions (es) and energy productivity (ye). es is the ratio of total energy consumption to carbon equivalent emissions, and ye is the ratio of GDP to energy consumption. Table 1 reports the descriptive statistics and stationary test results of the above variables. The panel unit root test mainly uses the LL and C test and the Fisher-ADF test. The former is the test under the same root condition, and the latter under different root conditions. The results show that the unit root tests of carbon productivity, effective investment rate, unit emission energy consumption, and energy productivity are significant, at least at the statistical level of 5%, which meets the modeling requirements of the panel data.

In addition, before the regression analysis, we further examined the spatial correlation of carbon productivity. Figure 1 draws the Moran scatter diagram of the carbon productivity of provinces and cities from 1995 to 2019 and reports the global Moran index and its test results. Throughout the study period, the global Moran’I over the years was significant at least at the 1% level, and the carbon productivity of provinces and cities showed indigenous spatial agglomeration characteristics. This suggests that factor flows and technical cooperation in neighboring provinces will indeed be more adequate and ignoring the spatial effects will lead to an estimation of the model and even wrong conclusions.

## 4. Empirical Results

### 4.1. Empirical Results of Cross-Sectional Data

Without considering the time effect and spatial effect of Equation (16), the estimation equation is degenerated into a cross-sectional regression model, which is similar to the cross-sectional setting in the traditional convergence literature [49]. In order to compare the effects of spatial effects, we first use different periods of interval data for cross-sectional ordinary least squares (OLS) regression: 1995–2010, 2010–2019, 1995–2019, and 2003–2019. Among them, 1995–2019 is the entire sample period, and 1995–2010 and 2010–2019 are its two sub-periods. This division considers that Chinese economy has entered a medium-low growth stage since 2011, while economic growth in the longer sub-period 2003–2019 is clearly driven by Chinese implementation of regional development strategies. For the entire sample period, we conducted a combined regression of mixed cross-sectional data at a four-year interval. The results are shown in Table 2. The number in the small brackets is the estimated robust standard error, and the convergence coefficients β of Equations (1)–(5) are highly indigenous at the 1% level. From 1995–2010 and 1995–2019, the carbon productivity of provinces and cities in China showed a convergence trend. The annual average convergence rates estimated by the single-section regression were 1.6% and 1.2%, respectively. The combined regression with four years as the interval also found similar observations. However, carbon productivity in the sub-periods of 2003–2019 and 2010–2019 showed divergent characteristics, with estimated annual convergence rates of −0.2% and −1.7%, respectively. It is necessary to point out that the above section regression estimation based on the traditional convergence model should be interpreted as absolute convergence because the influence of individual characteristics is not controlled in the equation.

Therefore, controlling the difference in the effective investment rate in the cross-sectional regression equation, we further estimate the conditional convergence of carbon productivity in each province. The results are shown in Table 3, and the corresponding period settings are the same as those in Table 2. The results show that the convergence coefficient is still highly indigenous at the 1% level, whether single-section or combined regression. The combined regressions for 1995–2010, 1995–2019, and four-year intervals estimated annual convergence rates of 3.0%, 1.5%, and 0.9%, respectively, significantly higher than the absolute convergence rates in Table 2. In addition, although the estimation results for 2010–2019 and 2003–2019 are still divergent, the convergence rate has also been significantly improved, indicating that ignoring the differences in individual characteristics, such as the effective investment rate, will lead to errors in the convergence estimation results. Similar situations also appear in the estimation results in Table 4.

Further introducing the spatial effect, Table 4 shows the QML estimation of the corresponding equation. Equations (1) to (5) can all be degenerated into the spatial error model (SEM) and pass the likelihood ratio (LR) test, indicating that the cross-sectional data constructed based on different periods tend to attribute the spillover effect to the spatial correlation of the error term. Compared with Table 3, these regressions produce similar estimation results. The spatial error term is statistically very significant, which preliminarily reflects the critical influence of the spatial effect contained in the theoretical model on the convergence of carbon productivity. Considering the spatial effect, the estimated conditional convergence rate improves significantly, and it also shifts from the previous divergence to convergence in the years from 2003–2019. It is worth noting that there is no conclusion that the effect of low-carbon technology spillover on convergence is greater than that of neutral technology spillover because the cross-sectional data based on interval structure omit time utility and cannot effectively identify the interaction between the spatial lag and time lag of carbon productivity, which is also fully verified in the estimation of panel data later.

### 4.2. Empirical Results of Panel Data

We further use the dynamic panel data method to estimate Equation (16). In order to overcome the impact of business cycles, the usual practice in the literature is to divide the sample into several shorter periods for regression with 5-year intervals [45,50]. However, our sample period is only 25 years. If a long period is adopted, many sample observations will be lost, which is not conducive to the effective identification of parameter estimation. Moreover, technology spillovers within China do not take years to occur relative to the spatial effects of cross-country panel data, as inter-regional factor flow and technical cooperation are less constrained. Therefore, based on the above considerations, this paper mainly uses the three years of dynamic panel data for regression analysis. Of course, we also further report the robustness test results based on the four-year span of the latter.

As mentioned above, when the lag dependent variable is specified on the right side of the regression equation, sufficient time-series observations should be included; otherwise, estimation bias of the dynamic panel data will occur. In order to ensure more effective convergence parameter estimation, we use the dynamic panel difference Gaussian Mixture Model (GMM) as the initial estimation and use the least squares dummy variable (LSDV) method for deviation correction estimation [51]. The results are shown in Table 5. Without spatial considerations, the average annual convergence rates for 1995–2019, 1995–2010, and 2010–2019 were 6.2%, 20.3%, and 4.3%, respectively, which were greater than the corresponding cross-sectional estimates of 3.0%, 3.7%, and−1.8% in Table 3. The higher estimation results are obtained by controlling the unobserved individual factors. The earlier convergence rate of carbon productivity is faster, consistent with the convergence results of Chinese carbon intensity estimated by the same method in the literature [52]. The convergence rate estimated in this paper is higher. The main reason is that we further add the identifiable individual effective investment rate factor to the right side of the equation. This variable also passes the statistical significance at least at the 10% level, showing a positive role in promoting the growth of carbon productivity. According to the 2010–2019 estimates, carbon productivity growth shifted from conditional divergence in cross-sectional regression to conditional convergence, indicating that the omission bias of cross-sectional regression was due to both individual factors and their spatial effects. Based on the sufficient sample size provided by the panel data, we further estimate the club convergence characteristics of Chinese carbon productivity. The division of Eastern, Central, and Western regions is consistent with the statistical norms published by NBS. The results show an aboriginal club convergence phenomenon in Chinese provincial carbon productivity. The β convergence coefficients in the Eastern, Central, and Western regions are highly aboriginal at the 1% level, and the annual convergence rates are 2.7%, 18.9%, and 7.4%, respectively.

Thethree years of dynamic panel data are still used, and Table 6 further reports the QML estimation results containing spatial factors. The SDPD estimation equation here is consistent with the previous theoretical model. We first carried out the Hausman test on it, and the results show that the individual fixed effect model is more effective at a higher statistical level. At the same time, the Davidson–MacKinnon test also rejects the general endogeneity problem of the model, and the QML estimation is robust and consistent. Specific estimates show, the spatial lag parameters of the two sub-periods and the entire sample period are positively statistically significant at the 1% level, indicating that carbon productivity and effective investment rate have a significant spatial correlation. At the same time, it can also be seen that the spatial effect is significant, whether it is synchronous lag or space–time lag. This means that spatial correlation should be considered in the regression of carbon productivity growth otherwise it may lead to omitted variable deviation. After considering spatial factors, the average annual convergence rates of 1995–2019 and 1995–2010 decreased to 3.7% and 13.4% compared with the corresponding model in Table 5, and the average annual convergence rate of 2010–2019 increased to 7.5%. This empirical conclusion is not inconsistent with the previous theoretical expectation but reflects the spatial spillover intensity differences between the two forms of technology at different development stages. Because the early Chinese economy had more extensive growth in rapid capital accumulation, knowledge spillover characterized by capital flow is stronger than the diffusion effect of technology diffusion, so the spatial technology spillover in this stage will delay the convergence rate. After 2010, Chinese economy has entered a stage of low- and medium-speed growth. The extensive input of elements has been further alleviated, and sustainable development characterized by structural transformation and technological innovation has become the dominant strategy. Therefore, during this period, the spatial spillover effect mainly manifests as technology diffusion. The late development advantages of less developed provinces are more fully reflected, and the overall convergence rate of carbon productivity is significantly accelerated. This conclusion is significant for promoting low-carbon technology diffusion to achieve intensive emission reduction policies.

From the results of regional regression in Table 6, the convergence coefficients of the Eastern, Central, and Western regions are statistically significant at 1%. The growth of carbon productivity in the Eastern, Central, and Western regions has prominent club convergence characteristics, and the convergence rate decreases in turn. Among them, the convergence rate of the Eastern region is 5.2%, which is higher than the 2.7% estimated by the non-spatial dynamic panel model, indicating that the low-carbon technology diffusion effect mainly characterizes the spatial spillover between the Eastern provinces and cities. The convergence rate in the Western region is 2.4%, lower than the 7.4% estimated by the non-spatial dynamic panel model, which means that the spatial spillover in the Western region mainly comes from the knowledge spillover effect of capital flow. The convergence rate in the Central region has not changed significantly, indicating an offset effect of spatial spillover between provinces and cities in the region. From the effective investment rate results, the local investment rate fails to pass the statistical test in most cases. On the contrary, its spatial interaction term has statistical significance in most cases. However, this is not necessarily interpreted as evidence that the local investment rate does not affect carbon productivity growth. It only shows that the spatial spillover of investment rate in adjacent areas contributes the most to the total effect [53].

### 4.3. Test of Convergence Mechanism

If carbon productivity can be decomposed into several significant economic drivers, the convergence factor effect can be further explored by combining the estimation equation [53]. So, according to the definition of carbon productivity, we perform the multiplicative decomposition of the following two factors:(18)$$y_{t}=\frac{Y_{t}}{C_{t}}=\frac{E_{t}}{C_{t}}\times \frac{Y_{t}}{E_{t}}=es_{t}\times ye_{t}$$
where es represents the energy structure effect, that is, the increase in energy consumption per unit of carbon equivalent emissions, it means the use of more low-carbon energy. ye is called energy efficiency, the reciprocal of which is the energy intensity effect index [54]. They are commonly used in the driving factor decomposition literature. The construction of these two indicators does not require additional data. The total energy consumption as a bridge can be easily achieved by decomposition. Due to $\ln y_{t}=\ln es_{t}+\ln ye_{t}$, the two indicators on the right side are brought into Equation (16) for spatial dynamic panel data regression. The factor effect of the convergence mechanism of carbon productivity can be further analyzed. To facilitate the comparison of convergence coefficients, we use a dynamic spatial panel model consistent with the corresponding equation in Table 6. Table 7 reports the estimation results of the whole country and the Eastern, Central, and Western regions.

The results in Table 7 show that these equations results are a good fit, and the spatial lag terms all passed the statistical test. Each province’s and city’s energy structure and energy efficiency have a significantly positive spatial correlation. This evidence shows that each province and city has maintained a relatively consistent policy on short-term energy structure adjustment and improvement of energy efficiency. The spatial–temporal lag parameters of energy structure lack statistical significance in the Central and Western region, which means that the provinces and cities in the region lack unity in the medium-term energy structure adjustment policies. On the contrary, the provinces and cities in the Eastern region maintain good medium-term policy consistency. Unlike the internal spatial spillover observed in a single area, the interaction and imitation between neighboring provinces and cities across the country are more adequate, so the time–space lag of the energy structure estimated by the overall sample is very significant. The spatial and temporal lag parameters of energy efficiency have obtained high statistical significance in the national, Eastern, and Western samples. The lack of statistical significance in the Central region shows that the medium-term policies of provinces and cities in the region in improving energy efficiency are not uniform, consistent with the estimation results of the corresponding carbon productivity convergence equation in Table 6. The β convergence coefficients of different estimation equations are highly significant at the 1% level, indicating that energy structure and energy efficiency affect the convergence of carbon productivity. From the national sample, the convergence rates of energy structure and energy efficiency are 1.6% and 5.4%, respectively, accelerating the convergence of carbon productivity. Regionally, energy structure and energy efficiency convergence rates were 11.3 percent and 9.4 percent in the East, −11.6 percent, and 19.9 percent in the Central, and −4.9 percent and 2.5 percent in the West, respectively. That is to say, the convergence of carbon productivity in the Eastern region comes from the convergence effect of two factors. The Central and Western regions mainly show the convergence effect of energy efficiency, and the energy structure shows the divergence effect. It can be seen that energy efficiency has become the main driving force for the convergence of carbon productivity in the whole country and even in various regions. The convergence of energy structure has only obtained evidence in the East, and the Central and West regions show indigenous divergence due to their internal energy endowment differences.

### 4.4. Robustness Test

To test whether the regression results are robust, we used 4-year interval data and 4-year average data to implement the spatial dynamic panel model. There is no technical difference between the four-year interval data structure and the previous three-year interval, but the two are different in time point selection. The average data are different from the time-point data, and the panel data are constructed based on the arithmetic average of each four adjacent years:$$\begin{array}{l} \ln y_{1}=\left(\ln y_{1995}+\ln y_{1996}+\ln y_{1997}+\ln y_{1998}\right)/4\\ \ln y_{2}=\left(\ln y_{1999}+\ln y_{2000}+\ln y_{2001}+\ln y_{2002}\right)/4\\ \vdots \\ \ln y_{6}=\left(\ln y_{2015}+\ln y_{2016}+\ln y_{2017}+\ln y_{2018}\right)/4 \end{array}$$

The estimation results of the spatial dynamic panel model of carbon productivity, energy structure, and energy efficiency are shown in Table 8. Using the panel data of four-year interval and four-year average, the estimated values of β convergence coefficient are 0.7323 and 0.9567, respectively. They have apparent statistical significance, indicating that the estimation of carbon productivity convergence with spatial effect is robust. In addition, the β convergence coefficients of energy structure and energy efficiency also have significant statistical significance, and both show energy efficiency convergence is the dominant force of Chinese carbon productivity spatial convergence, which is basically consistent with the corresponding convergence mechanism estimation results in Table 7. It is worth noting that the mean panel estimation results reveal the divergence trend of energy structure, indicating that the positive effect of energy structure adjustment on carbon productivity convergence is small and lacks stability. Finally, using the three-year interval panel data, we further give the estimation results that the spatial weight matrix has more neighborhoods, that is, the reciprocal of the longitude and latitude distance of the geographical centers of each province and city is used as the weight and the treatment is standardized, which means that the spatial effect of technology spillover decreases with the increase in geographical distance. As can be seen from the estimation results of Table 8, the beta convergence coefficients of carbon productivity, energy structure, and energy efficiency are very obvious and have similar convergence characteristics and factor effects with adjacent weight matrix estimation, which further shows that the spatial convergence trend of carbon productivity in China is robust.

### 4.5. Further Discussion

Based on the background that the current research on the convergence of carbon emissions and energy consumption lacks theoretical guidance [10,40], this paper constructs a Solow model containing two kinds of spatial technology spillover effects and proposes that technology diffusion can promote the convergence of carbon productivity. In contrast, knowledge spillover will inhibit the confluence, consistent with the different inferences of the spatial effects of two kinds of technology spillover in the literature [47,50]. Different from the existing literature on spatial spillover effects can accelerate the convergence of carbon intensity estimation results [52], this paper uses the estimation results of Chinese provincial dynamic spatial panel data to obtain different empirical conclusions and better support the theoretical expectation of spatial technology spillover difference because the convergence rate of carbon productivity depends on the comparison of the two spatial technology spillover forces. Nevertheless, we still cautiously give possible explanations for the differences in results. On the one hand, this paper further controls the space–time lag term and the adequate investment rate difference in the estimation equation, a significant control variable in the neoclassical framework containing spatial effects and has an essential impact on the convergence of carbon productivity. On the other hand, this paper’s spatial interaction term and spatial–temporal interaction term have better statistical performance in the dynamic panel model, so the collinearity effect from the spatial lag term and spatial–temporal lag term may be more negligible. In addition, the Chinese government established a formal system to constrain energy intensity around 2005, and the short-term energy intensity mitigation measures adopted by provinces and cities are relatively consistent with the medium-term sustainable development strategy, which explains the statistical evidence of the spatial correlation between the spatial lag term and the spatial–temporal lag term in the same period. It is worth noting that the spatial uniformity of energy intensity policies has not received statistical evidence of club convergence in the Central region, which may be because the Central region is adjacent to the East and West regions at the same time and that emission reduction strategies in provinces and cities of different geographical locations are also affected by extraterritorial areas.

In addition, further convergence mechanism tests show that the development of energy efficiency dominates the spatial transfer of technology, so the overall convergence of carbon productivity in China mainly comes from the apparent convergence of energy efficiency in provinces and cities, which is consistent with the view in the literature [55]. The impact of energy structure on carbon emissions is crucial in theory, and the existing literature also provides empirical evidence for China [56]. This paper proves that the apparent divergence of energy structure effect among provinces becomes the constraint factor of carbon productivity convergence, mainly manifested in the inconsistency of energy structure adjustment in the Central and West provinces and the slow and unstable adjustment process. On the one hand, most provinces and cities have long been dominated by coal consumption and low-carbon energy structure adjustment has only accelerated in recent years. On the other hand, due to the impact of regional energy endowments, the provinces and cities in energy structure adjustment also failed to maintain the relative consistency of policy implementation. The low-carbon process of developed provinces is relatively fast, while the Central and West provinces are relatively slow.

A potential value of this paper is that it has important implications for global emission reduction actions and internal policies of a country, because the spatial knowledge spillover carried by capital cross-border flows is not conducive to the pursuit of carbon productivity in backward countries or regions, and even delays the overall convergence rate. The literature evidence of ‘pollution haven’ hypothesis provides a partial explanation [57,58], while the opposite empirical conclusion also shows the threshold effect evidence of a nonlinear relationship [59]. In other words, the knowledge spillover effect relying solely on capital flow does not necessarily promote the overall emission reduction of the world and a country. The current contribution of independent action is not sufficient to achieve the commitment of the Paris Agreement. On the contrary, the diffusion of low-carbon technology can significantly reduce carbon equivalent input in the production process, thereby speeding up the catch-up speed of backward countries or regions, which can significantly release the emission reduction potential of these economies. Emission reduction actions on a global scale and even within a country cannot be at the expense of the development rights of developing countries or regions. Sustainable emission reduction actions should focus on disseminating and diffusion of low-carbon technologies. Reducing technological monopoly and institutional barriers is the key to realizing global emission reduction goals. On the other hand, low-carbon projects supported by developed countries or regions should strictly control the ‘pollution haven’ effect and prevent the restrictions of high-carbon capital flows on the sustainable growth of developing countries or regions.

## 5. Conclusions

The contribution of this paper is to introduce carbon equivalent elements into the neoclassical model, propose a carbon productivity convergence model from the perspective of sustainable growth, and prove the different effects of two spatial technology spillover forms on convergence. Using Chinese provincial panel data from 1995 to 2019, we tested the convergence hypothesis of carbon productivity. Our empirical conclusions are consistent with theoretical expectations and further explore the effect of energy structure and energy efficiency on convergence. The results show that the spatial dynamic panel data can effectively control the variable deviation. The spatial spillover effect also shows no simple correlation between individual factors and low carbon growth. The convergence rate of dynamic panel regression estimation is significantly higher than that of cross-sectional regression. Whether the convergence rate of spatial panel regression is greater than that of non-spatial panel regression depends on the spatial spillover intensity difference of the two technologies.

Over time, provincial differences in China’ s carbon productivity have narrowed. The statistically significant spatial lag coefficient and spatial–temporal lag coefficient show that although provinces and cities are tending to a unique steady-state equilibrium, they do not do so independently, but tend to show movement similar to that of neighboring neighbors. Therefore, in most cases, the short-term and medium-term measures taken by provinces and cities to improve carbon productivity are relatively consistent. In the early stage, spatial dependence is mainly manifested as knowledge spillover with capital flow as the carrier, and this spatial technology spillover has a delaying effect on the convergence rate of a closed economy under the neoclassical framework. In the past decade, the rapid accumulation of capital in various provinces and cities has been alleviated. The spread and diffusion of low-carbon technology has become the dominant spatial spillover form of the sustainable growth strategy. The convergence rate of carbon productivity is significantly faster than that of the non-spatial dependence economy. Nonetheless, Chinese early carbon productivity convergence is still much faster than it was in the current period, largely due to a phased shift in the country’s overall slowdown since 2010.

Of course, this study also has some limitations. Firstly, in choosing a spatial econometric model, we failed to further propose the estimation method of the dynamic panel SEM model, and the study of extending the static SEM model to a dynamic model will become significant. Second, the carbon emission estimation method in this paper has certain limitations. The emission factors will change significantly under different technical levels and production conditions. Although we used the updated provincial total energy data to adjust, this apparent method may not fully reflect the regional energy technology differences. Additionally, the emission factor method is a theoretical calculation method, and there is a significant deviation between the theoretical and actual values. This may cause some variations in identifying spatial spillover effects of carbon productivity. However, our findings also provide evidence of spatial spillover differences.

## Figures and Tables

**Figure 1 ijerph-19-04606-f001:**
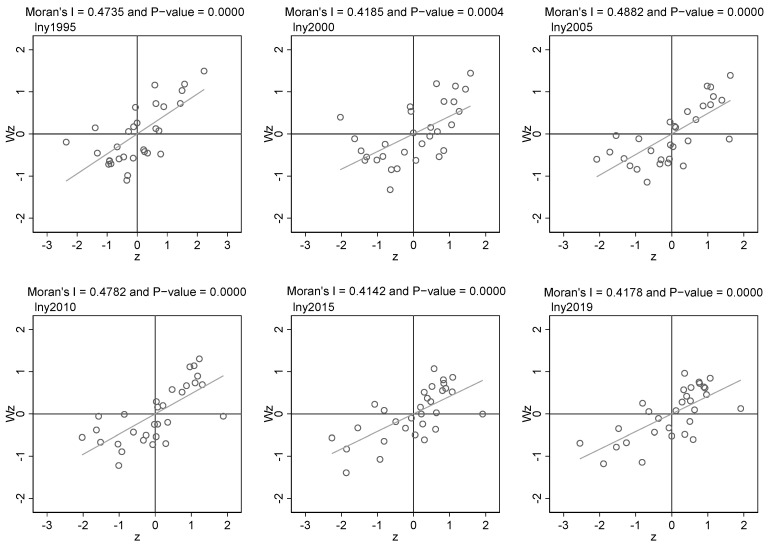
Moran scatter plot of provincial carbon productivity in China.

**Table 1 ijerph-19-04606-t001:** Variable Description and Stability Test (1995–2019).

Variable		Mean	Std. Dev.	Min	Max	Obs.	LL and C	Fisher-ADF
lny	overall	0.450	0.517	−1.087	1.970	750	−2.45 *** (0.007)	9.54 *** (0.000)
(carbon productivity)	between		0.409	−0.335	1.095	30		
	within		0.324	−0.438	1.332	25		
lns	overall	1.216	0.509	0.105	2.611	750	−2.69 *** (0.004)	8.62 *** (0.000)
(Effective investment rate)	between		0.177	0.793	1.601	30		
	within		0.478	0.061	2.656	25		
lnes	overall	0.465	0.038	0.421	0.702	750	−1.81 ** (0.036)	3.92 *** (0.000)
(Unit emission energy consumption)	between		0.033	0.424	0.562	30		
	within		0.020	0.324	0.624	25		
lnye	overall	−0.015	0.501	−1.507	1.269	750	−3.89 *** (0.000)	10.00 *** (0.000)
(energy productivity)	between		0.398	−0.777	0.553	30		
	within		0.313	−0.895	0.859	25		

Note: *** and **, are significant at the 1% and 5%levels, respectively; *p*-values are in parentheses for parameters.

**Table 2 ijerph-19-04606-t002:** Absolute Convergence Test of Cross-Sectional Data.

Equation	(1)	(2)	(3)	(4)	(5)
	1995–2010	2010–2019	1995–2019	2003–2019	Pooled
lny_0_	0.7493 ***	1.1616 ***	0.7319 ***	1.0287 ***	0.9759 ***
	(0.0853)	(0.1050)	(0.1438)	(0.1516)	(0.0261)
Constant	0.5356 ***	0.4256 ***	1.0385 ***	0.6681 ***	0.1847 ***
	(0.0449)	(0.0724)	(0.074)	(0.0813)	(0.0181)
Implied λ	0.0192	−0.0166	0.0130	−0.0018	0.0061
β = e^−^^λt^	t = 15	t = 9	t = 24	t = 16	t = 4
R-square	0.6352	0.843	0.3786	0.6963	0.8903
*n*	30	30	30	30	180

Note: *** is significant at the 1% levels, respectively; robust standard errors are in parentheses for parameters.

**Table 3 ijerph-19-04606-t003:** Conditional Convergence Test of Cross-Sectional Data.

Equation	(1)	(2)	(3)	(4)	(5)
	1995–2010	2010–2019	1995–2019	2003–2019	Pooled
lny_0_	0.6378 ***	1.1914 ***	0.7014 ***	1.0128 ***	0.9661 ***
	(0.0909)	(0.1163)	(0.1360)	(0.1494)	(0.0272)
lns	−0.3793 *	0.0771	−0.1502	−0.0607	0.0590 ***
	(0.2122)	(0.0645)	(0.1499)	(0.1136)	(0.0223)
Constant	1.0111 ***	0.2685	1.3143 ***	0.7854 ***	0.1116 ***
	(0.2575)	(0.1592)	(0.2865)	(0.2190)	(0.0350)
Implied λ	0.0300	−0.0195	0.0148	−0.0008	0.0086
β = e^−^^λ^^t^	t = 15	t = 9	t = 24	t = 16	t = 4
R-square	0.6724	0.8474	0.3966	0.6991	0.8936

Note: *** and * are significant at the 1% and 10% levels, respectively; robust standard errors are in parentheses for parameters.

**Table 4 ijerph-19-04606-t004:** Conditional Convergence Test of Spatial Cross-Sectional Data.

Equation	(1)	(2)	(3)	(4)	(5)
	1995–2010	2010–2019	1995–2019	2003–2019	Pooled
lny_0_	0.5709 ***	1.1716 ***	0.4932 **	0.8942 ***	0.8819 ***
	(0.1261)	(0.1114)	(0.1962)	(0.1442)	(0.0435)
lns	−0.2281	0.0251	−0.1146	−0.0877	−0.0128
	(0.1798)	(0.0613)	(0.1177)	(0.0860)	(0.0319)
Constant	0.8071 ***	0.3226	1.1635 ***	0.8165 ***	0.2948 ***
	(0.3070)	(0.2140)	(0.3958)	(0.2902)	(0.0670)
w*e	0.8134 ***	0.8182 ***	0.8213 ***	0.8182 ***	0.6089 ***
	(0.0961)	(0.0943)	(0.0933)	(0.0944)	(0.0805)
Implied λ	0.0374	−0.0176	0.0295	0.0070	0.0314
β = e^−^^λt^	t = 15	t = 9	t = 24	t = 16	t = 4
Pseudo R^2^	0.6692	0.8454	0.3965	0.698	0.8667
*n*	30	30	30	30	180
LR_SLM	5.77 *	20.97 ***	1.45	8.37 **	52.06 ***
LR_SEM	0.30	1.12	0.17	0.66	0.11

Note: ***, **, and * are significant at the 1%, 5%, and 10% levels, respectively; robust standard errors are in parentheses for parameters.

**Table 5 ijerph-19-04606-t005:** Conditional Convergence Test of Dynamic Panel Data.

Equation	(1)	(2)	(3)	(4)	(5)	(6)
	1995–2019	1995–2010	2010–2019	East	Central	West
L.lny	0.8299 ***	0.5433 ***	0.8779 ***	0.9210 ***	0.5675 ***	0.8021 ***
	(0.0381)	(0.0486)	(0.0573)	(0.0392)	(0.0774)	(0.0724)
lns	0.1201 ***	0.1255 **	0.0712 **	0.0882 **	0.2604 ***	0.1374 ***
	(0.0224)	(0.0511)	(0.0325)	(0.036)	(0.0576)	(0.0409)
Implied λ	0.0622	0.2034	0.0434	0.0274	0.1888	0.0735
β = e^−^^λt^	t = 3	t = 3	t = 3	t = 3	t = 3	t = 3
*n*	240	150	90	88	64	88

Note: *** and ** are significant at the 1% and 5% levels, respectively; robust standard errors are in parentheses for parameters.

**Table 6 ijerph-19-04606-t006:** Conditional Convergence Test of Spatial Dynamic Panel Data.

Equation	(1)	(2)	(3)	(4)	(5)	(6)
	1995–2019	1995–2010	2010–2019	East	Central	West
L.lny	0.8944 ***	0.6693 ***	0.7975 ***	0.8562 ***	0.5678 ***	0.9302 ***
	(0.0605)	(0.0772)	(0.0756)	(0.051)	(0.1250)	(0.070)
w*lny	0.5754 ***	0.4194 ***	0.5341 ***	0.4647 ***	0.3195 ***	0.5656 ***
	(0.0478)	(0.0882)	(0.0793)	(0.0973)	(0.0832)	(0.0398)
L.w*lny	−0.5942 ***	−0.4539 ***	−0.4341 ***	−0.4380 ***	−0.2530	−0.5250 ***
	(0.0813)	(0.1155)	(0.1097)	(0.0928)	(0.1689)	(0.1280)
lns	−0.0153	−0.1198 *	0.0335	0.0103	0.0810	0.0334
	(0.0290)	(0.0618)	(0.0224)	(0.0420)	(0.0721)	(0.0470)
w*lns	0.1028 **	0.2810 ***		0.0944 **	0.1442 *	
	(0.0419)	(0.0433)		(0.0435)	(0.0767)	
Implied λ	0.0372	0.1338	0.0754	0.0518	0.1887	0.0241
β = e^−^^λt^	t = 3	t = 3	t = 3	t = 3	t = 3	t = 3
R-square	0.9289	0.8193	0.9564	0.9468	0.8980	0.9046
Hausman	31.77 ***	36.87 ***	9.26 *	23.18 ***	16.68 ***	18.76 ***
D-M test	2.06	1.22	1.36	1.74	0.06	2.37
LR_SLM	14.26 ***	21.74 ***	2.69	11.59 ***	3.68 *	1.95
*n*	240	150	120	88	64	88

Note: ***, **, and * are significant at the 1%, 5%, and 10% levels, respectively; robust standard errors are in parentheses for parameters.

**Table 7 ijerph-19-04606-t007:** Test of Convergence Mechanism for Spatial Dynamic Panel Data.

Equation	Total		East		Central		West	
	(1) lnes	(2) lnye	(3) lnes	(4) lnye	(5) lnes	(6) lnye	(7) lnes	(8) lnye
L.lnes	0.9543 ***		0.7118 **		1.4173 ***		1.1576 ***	
	(0.0476)		(0.2896)		(0.1899)		(0.1138)	
L.w*lnes	0.4812 ***		0.3638 **		−0.0564		−0.2797	
	(0.1293)		(0.1781)		(0.0996)		(0.1953)	
L.lnye		0.8499 ***		0.7554 ***		0.5510 ***		0.9288 ***
		(0.0397)		(0.0433)		(0.1246)		(0.0659)
L.w*lnye		−0.5532 ***		−0.3429 ***		−0.2331		−0.5390 ***
		(0.0583)		(0.0927)		(0.1689)		(0.1257)
lns	−0.0070 **	−0.0082	−0.0219	0.0086	0.0025	0.0799	0.0019	0.0352
	(0.0033)	(0.0227)	(0.0134)	(0.0395)	(0.0028)	(0.0711)	(0.0032)	(0.0464)
w*lns	0.0028	0.0991 ***	0.0231	0.1033 ***	−0.0020	0.1391 *		
	(0.0036)	(0.0341)	(0.0177)	(0.0376)	(0.0036)	(0.0734)		
rho	0.1700 *	0.5662 ***	0.1825 **	0.4504 ***	0.1726 ***	0.3220 ***	0.1508 *	0.5730 ***
	(0.0896)	(0.0624)	(0.0772)	(0.1049)	(0.0468)	(0.0815)	(0.0774)	(0.0389)
Implied λ	0.0156	0.0542	0.1133	0.0935	−0.1163	0.1987	−0.0488	0.0246
β = e^−^^λt^	t = 3	t = 3	t = 3	t = 3	t = 3	t = 3	t = 3	t = 3
R-square	0.8559	0.9217	0.7696	0.9332	0.9031	0.894	0.8833	0.9073
*n*	240	240	88	88	64	64	88	88

Note: ***, **, and * are significant at the 1%, 5%, and 10% levels, respectively; robust standard errors are in parentheses for parameters.

**Table 8 ijerph-19-04606-t008:** Robustness Test Results Based on SDM Model.

Equation	4-Year Interval			4-Year Average			Distance Weight		
	(1) lny	(2) lnes	(3) lnye	(4) lny	(5) lnes	(6) lnye	(7) lny	(8) lnes	(9) lnye
L.lny	0.7323 ***			0.9567 ***			0.9653 ***		
	(0.1082)			(0.0845)			(0.0499)		
L.w*lny	−0.3742 ***			−0.5955 ***			−0.7548 ***		
	(0.1265)			(0.0778)			(0.0845)		
L.lnes		0.8597 ***			1.3820 ***			0.9959 ***	
		(0.2607)			(0.2942)			(0.2621)	
L.w*lnes		0.6861 ***			0.1992			−0.1000	
		(0.2502)			(0.2879)			(0.2180)	
L.lnye			0.6855 ***			0.8996 ***			0.9125 ***
			(0.1007)			(0.0760)			(0.0468)
L.w*lnye			−0.3358 ***			−0.5596 ***			−0.7063 ***
			(0.1214)			(0.0740)			(0.0871)
lns	0.0195	−0.0103	0.0302	0.0183	−0.0035	0.0283	−0.0069	−0.0032	−0.0038
	(0.0470)	(0.0108)	(0.0403)	(0.0469)	(0.0140)	(0.0409)	(0.0253)	(0.0071)	(0.0215)
w*lns	0.1673 **	0.0072	0.1568 **	−0.0225	−0.0006	−0.0167	0.1852 ***	0.0026	0.1889 ***
	(0.0661)	(0.0120)	(0.0625)	(0.0720)	(0.0143)	(0.0697)	(0.0479)	(0.0095)	(0.0440)
rho	0.3851 ***	0.1557 *	0.3785 ***	0.7253 ***	0.2630 **	0.7114 ***	0.5432 ***	0.2829 ***	0.5258 ***
	(0.0682)	(0.0886)	(0.0694)	(0.043	(0.1119)	(0.0456)	(0.0957)	(0.0851)	(0.0968)
Implied λ	0.0779	0.0378	0.0944	0.0111	−0.0809	0.0265	0.0118	0.0014	0.0305
β = e^−λt^	t = 4	t = 4	t = 4	t = 4	t = 4	t = 4	t = 3	t = 3	t = 3
R-square	0.8515	0.7865	0.8381	0.9004	0.8549	0.9078	0.9560	0.8811	0.9521
*n*	180	180	180	150	150	150	240	240	240

Note: ***, **, and * are significant at the 1%, 5%, and 10% levels, respectively; robust standard errors are in parentheses for parameters.

## Data Availability

Not applicable.

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
