# Peer review of "Spatial Convergence of Carbon Productivity: Theoretical Analysis and Chinese Experience"

_ijerph, 2022, doi:10.3390/ijerph19084606_

Round 1

Reviewer 1 Report

The paper is interesting as it deals with a topical issue for a very relevant country in terms of GHG emissions. The quality of the presentation is good and also the scientific soundness is adequate. I have only two major concerns dealing with:

  • lines 49- 50. The authors state that “To reduce carbon dioxide emissions, it is mainly to improve carbon productivity.” Indeed, this sentence could be misleading. In fact, the carbon productivity of one unit (country or firm) can improve even if the GHG emissions increase and the GDP increases at a higher rate. Thus, increasing carbon productivity does not mean reducing GHG emissions. As stated by other authors, indeed, to ensure that GHG emissions are decreasing relative GHG measures (like carbon productivity) should be accompanied by absolute GHG emissions trend analysis. Besides this, there may also be a negative relation between the increase in resource efficiency and the absolute reduction of resource use. In fact, according to the so-called Jevons’ Paradox or “rebound effect”, the overall effect of improved resource efficiency can be an increase in the use of the resource itself. The magnitude of this rebound effect is driven by the degree of substitution between factors of production and is still a question of debate in the energy sector.

I would suggest the authors to better discuss these issues, also in the discussion section by referring to other studies in this field. Among others:

  • Coderoni S., Vanino S. (2022), The farm-by-farm relationship among carbon productivity and economic performance of agriculture, Science of the Total Environment 819 (2022) 153103, http://dx.doi.org/10.1016/j.scitotenv.2022.153103
  • Siami, N., Winter, R.A., 2021. Jevons’ paradox revisited: Implications for climate change. Econ. Lett. 206, 109955. https://doi.org/10.1016/j.econlet.2021.109955.
  • Sorrell, S., 2009. Jevons’ Paradox revisited: The evidence for backfire from improved energy efficiency. Energy Policy 37, 1456–1469. https://doi.org/10.1016/j.enpol.2008.12.003.
  • The methodology used to derive GHG emissions is based on an IPCC tier 1 approach. This methodology is based on a linear relationship between activity data and emission factors; thus it is not able to capture eventual major carbon-efficient energy production technologies (e.g. renewable energy production). A tier 2 methodology would have been preferred. The authors should discuss the shortcomings of the methodology used to calculate GHG emission and eventually propose some solutions to obtain more reliable estimations.

Minor comments:

To ease the reading of the manuscript I would suggest writing in the results table the names of the variables instead of the coefficients.

Line 352-353 the sentence seems to be incomplete.

Author Response

Dear reviewer:

Thanks for the reviewer’ comments concerning our manuscript entitled “Carbon Productivity Convergence Hypothesis: a Spatial Solow Model” (No. ijerph-1635307). Those comments and suggestions are helpful for revising and improving our paper. We have carefully made corrections and revised the manuscript which we wish to be approved. The modified content is marked by “Track Changes”. Please see the attachment for details.

Once again, thank you very much for your comments and suggestions

Reviewer 2 Report

The content contained is suitable for publication after the following comments have been corrected:
- the title can be changed so that it does not include the word "hypothesis" and that the study was based on data from China;
- Line 53-59 - maybe it is better to present the content in the form of problem analysis (as the authors later call them) instead of questions, unless these are research questions?
- Model symbols in the text of the article should be corrected, eg lines 149-160;
- Line 164: authors write We can write formula (5)… .. in matrix form, but I couldn't see a matrix;
- Line 258, 290 on so on: strange underline under the text;
- Line 292: open bracket, move it down;
- Table 2: illegible column name;
- Pattern number 25 is not centered
- Empty lines between lines 576 and 577
The discussion of the results and conclusions should be written separately.

Author Response

(The authors gave the same response as above.)

Reviewer 3 Report

  1.  Section 3.3 "Data declaration" change it to data and variables description.
  2. Line 314,  "Carbon productivity is calculated by the ratio of GDP to carbon equivalent emissions, and the GDP of each province is adjusted to the comparable price in 2010" there is no justification provided why the year 2010 take as the base year for price.
  3. Table 2, Column 5 heading is in the Chinese language, translate into Th English language.
  4.  There is no justification provided that why your applied method is so primitive to other methods?
  5.  Some abbreviations are missing such as GMM etc, check the whole draft regarding this issue. 

Author Response

(The authors gave the same response as above.)

Reviewer 4 Report

The paper proposes to investigate the convergence hypothesis of carbon productivity under sustainable growth and prove the different effects of knowledge spillover technology diffusion on convergence. The study considers China’s provincial spatial dynamic panel data from 1995 to 2019. The results show that China's s provincial carbon productivity has conditional convergence and club convergence characteristics.

The paper also provides cross-sectional and panel data analysis, and, in this further analysis, it finds convergence speed of dynamic panel regression estimation is greater than that of cross-sectional one.

I find this work is interesting. Carbon productivity is an important point in the study of global warming and, this topic is relevant with the discussions in place at the main international institutions working on climate changes and remedies.

Even if the topic is appealing, I think the paper suffers from some important shortcomings referred to in the data.

My first concern is related to the missing table with a summary variable statistic. To understand the paper and the methodology used it is relevant to add as much information as possible.

My second point refers to the description of the estimator authors are using. Is it a Fixed effects estimator? If the answer is yes, some problems arise. Indeed, for running a GMM estimation some specific rules should be applied. Here we can observe a large T and a small N: thus, the number of observations included in the panel structure analysis by province is not enough for providing robust analysis. The results will be valid, but there may be an issue with inference with small N. This is summarized by Wooldridge (2012, p.490)

“When T is large, and especially when N is not very large (for example, N = 20 and T = 30), we must exercise caution in using the fixed effects estimator. Although exact distributional results hold for any N and T under the classical fixed effects assumptions, inference can be very sensitive to violations of the assumptions when N is small, and T is large. In particular, if we are using unit root processes—see Chapter 11—the spurious regression problem can arise. First differencing has the advantage of turning an integrated time-series process into a weakly dependent process. Therefore, if we apply first differencing, we can appeal to the central limit theorem even in cases where T is larger than N. Normality in the idiosyncratic errors is not needed, and heteroskedasticity and serial correlation can be dealt with as we touched on in Chapter 13. Inference with the fixed effects estimator is potentially more sensitive to nonnormality, heteroskedasticity, and serial correlation in the idiosyncratic errors”

Furthermore, if the authors are using large-T-small-N asymptotics then, the authors should consider problems connected to the stationarity, cointegration. Do the author account for it?

Do the authors' test for endogeneity problems?

Author Response

(The authors gave the same response as above.)

Reviewer 5 Report

Dear Authors,

The submitted paper proposed the convergence hypothesis of carbon productivity under sustainable growth based on the neoclassical framework, and prove the different effects of knowledge spillover and technology diffusion on convergence in the context of China. The method and model are applicable and the conclusions are of practical significance for developing countries to explore the path of energy conservation and emission reduction.

Some empirical issues need to be improved.

  1. Descriptive statistics and unit root tests should be supplemented.
  2. Why choose the SDM rather than SLM and SEM?Several indicators and tests results used for spatial econometric model selection should be reported.
  3. Does the SDM estimation use random effects or fixed effects? If it is the fixed effect, is it space-time fixed or space-fixed?

Some problems of presentation need to be improved.

  1. For instance, Page 1 line37-38 The subject of the sentence is not clear.
  2. Page 2 Line 49-50 Improving carbon productivity is not a sufficient condition for reducing carbon emissions.
  3. Page 2 Line 72-74 It should be practical significance rather than theoretical.
  4. Page 2-3 Literature review over describes economic convergence, environmental convergence and other topics, and lacks a detailed review of the literature on carbon emission convergence, which is closely related to the topic of this paper.
  5. Page 19-20 The conclusions are only the summary of empirical results, and lack the description of the innovation and practical value of this paper. In addition, the research limitations also need to be supplemented accordingly.

Some formatting issues need to be improved.

  1. For example, Page 11 Line 354 Chinese characters appear in Table 2.
  2. Page 17&18 It should be Table 7&8 instead of Table 1&2.

Author Response

Dear reviewer:

Thanks for the reviewer’ comments concerning our manuscript entitled “Carbon Productivity Convergence Hypothesis: a Spatial Solow Model” (No. ijerph-1635307). Those comments and suggestions are helpful for revising and improving our paper. We have carefully made corrections and revised the manuscript which we wish to be approved. The modified content is marked by “Track Changes”. Please see the attachment for details.

Once again, thank you very much for your comments and suggestions.

Round 2

Reviewer 1 Report

The paper has improved significantly.